# Late initiation of antenatal care visit amid implementation of new antenatal care model in Sub-Saharan African countries: A multilevel analysis of multination population survey data

**Kusse Urmale Mare**[1]*, **Gashaye Gobena Andargie**[1], **Abdulkerim Hassen Moloro**[1],
**Ahmed Adem Mohammed**[1], **Osman Ahmed Mohammed**[1], **Beriso Furo Wengoro**[2],
**Begetayinoral Kussia Lahole**[3], **Tesfahun Simon Hadaro**[3], **Simeon Meskele Leyto**[4],
**Petros Orkaido Mamo**[5], **Abdulhakim Hora Hedato**[6], **Beminate Lemma Seifu**[6], **Temesgen
Gebeyehu Wondmeneh**[1], **Oumer Abdulkadir Ebrahim**[6], **Kebede Gemeda Sabo**[1]

1 Department of Nursing, College of Medicine and Health Sciences, Samara University, Samara, Ethiopia,
2 Department of Biomedical Sciences, College of Medicine and Health Sciences, Samara University,
Samara, Ethiopia, 3 Department of Midwifery, College of Medicine and Health Sciences, Arba Minch
University, Arba Minch, Ethiopia, 4 Department of Biomedical Sciences, College of Medicine and Health
Sciences, Wachemo University, Hossana, Ethiopia, 5 Clinical Governance and Quality Improvement
Department, Karat Primary Hospital, Konso, Ethiopia, 6 Department of Public Health, College of Medicine
and Health Sciences, Samara University, Samara, Ethiopia

* kussesinbo@gmail.com

University, Faculty of Medicine, AFGHANISTAN

**Data Availability Statement:** The dataset used and
analyzed in this study can be accessed from the

## Abstract

### Introduction

Although late initiation of antenatal care has been linked with adverse pregnancy outcomes,
a significant number of pregnant women in resource-limited regions are seeking this care
late. There has been a lack of information on the extent and factors influencing late initiation
of antenatal care in the African context following the implementation of the new antenatal
care model in 2016. Thus, we aimed to determine the pooled prevalence of late antenatal
care visit and its determinants among women in Sub-Saharan Africa (SSA) using national
surveys conducted after the implementation of new guidelines.

### Methods

We analyzed data from the recent demographic and health survey (DHS) conducted in 16
SSA between 2018 and 2022 using a weighted sample of 101,983 women who had antena-
tal care follow-up during their index pregnancy. A multilevel logistic regression models were
fitted and likelihood and deviance values were used for model selection. In the regression
model, we used adjusted odds ratios along with their corresponding 95% confidence inter-
vals to determine the factors associated with late antenatal care visit.

### Results

The pooled prevalence of late antenatal care visit among pregnant women in SSA was
53.8% [95% CI: 46.2% - 61.3%], ranging from 27.8% in Liberia to 75.8% in Nigeria.

DHS website (https://dhsprogram.com/data/available-datasets.cfm).

**Funding:** The author(s) received no specific funding for this work.

**Competing interests:** The authors have declared that no competing interests exist.

**Abbreviations:** ANC, Antenatal Care; AOR, Adjusted Odds Ratio; COR, Crude Odds Ratio; DHS, Demographic and Health Survey; LL, Log-likelihood; SSA, Sub-Saharan Africa; WHO, World Health Organization.

Women's age and education, working status, partner's education, health insurance coverage, birth order, household wealth, age at marriage, decision on health care, residence, and community-level women's illiteracy were the factors associated with late antenatal care visit.

## Conclusion

More than half of pregnant women in SSA initiated attending antenatal care during late pregnancy, with significant differences seen among countries. These results underscore the necessity for focused interventions to tackle these issues and enhance prompt access to antenatal services for better maternal and child health outcomes in the area. Moreover, creating tailored interventions for younger women, those with multiple previous births, employed women, and those who experienced early marriage to address their specific challenges and obstacles in seeking care is crucial.

## Introduction

Timely antenatal care (ANC) visit is crucial for early identification, management, and prevention of health conditions that can negatively affect the well-being of mother and their unborn children [1, 2]. World Health Organization (WHO) recommends that women should begin attending ANC services during the first trimester and initiation of a care after this timeframe is deemed late [1, 3].

Studies have shown that delayed initiation of ANC is associated with negative maternal and newborn outcomes. For example, it has been found that late ANC visit is linked with a higher risk of maternal anemia [4], low birth weight [4–9], fetal and neonatal death [7, 10], preterm birth [8, 10], stillbirth [9], and restricted fetal growth [10].

The prevalence of late ANC visit varies significantly across different regions of the globe. In Ethiopia, for instance, the proportion of women who begin ANC late ranges from 32% [11] to 66% [12]. Similarly, studies in different African countries reveal that the rate of late ANC initiation was 50% in Kenya [13], 85.3% in Tanzania [14], and 82.4% in the Democratic Republic of Congo [15]. Additionally, other studies indicated the level of late visit to be 73% in Uganda [16], 61% in Rwanda [17], and 86% in Bangladesh [18].

Several factors have been identified as contributing to delayed ANC visit. Sociodemographic variables such as age [12, 15, 19, 20], residency [11, 13], marital status [11, 17, 19], education [12–15, 21], occupation [20], income [11, 13], family size [13, 20], partner cohabitation [15, 16], distance to health facility [15–17, 20–22], and health insurance [17] have been associated with late initiation of ANC. Moreover, other factors like birth order [11, 13, 14, 17], unplanned pregnancy [12, 17, 19, 21], prior institutional delivery [16], previous ANC service use [19], knowledge of ANC [21], and misconception about the timing of visits [19] have also been linked to late visit.

Globally, various interventions have been put in place to improve maternal and newborn health, particularly in reducing the rates of late initiation of ANC. One notable initiative was the launch of a new ANC model by the WHO in 2016 [2], which increased the minimum number of ANC contacts from four visits in the former recommendation to eight visits. This change aimed at enhancing opportunities for early detection and management of potential health risks. Additionally, comprehensive interventions have been taken at both health system and community levels to improve the quality and accessibility of the services, as well as to

promote community awareness on timely initiation of ANC [23–26]. These efforts have been primarily focused on improving the early initiation of ANC, particularly in resource-limited settings [26, 27].

Although late initiation of ANC is associated with adverse pregnancy and childbirth outcomes [5, 6, 9, 10], a significant number of women in resource-limited regions like Africa initiate receiving this care during late pregnancy [12–14, 16, 21]. To the authors' knowledge, there is limited evidence regarding the prevalence of late ANC visit and its determinants in Sub-Saharan Africa (SSA) following the implementation of the 2016 new antenatal care model. Understanding the level and the factors influencing late ANC visits is important for improving compliance with the new ANC recommendations. Therefore, this study aims to estimate the pooled prevalence of late ANC visit and its determinants among pregnant women in SSA.

## Methods

### Data source and participants

This study utilized data from the recent demographic and health survey (DHS) conducted in 16 SSA between 2018 and 2022. The selection of these countries was based on the availability of a standardized and unrestricted DHS dataset that contained the outcome and necessary explanatory variables. DHS is a nationally representative survey conducted every five years to collect data on basic sociodemographic characteristics and various health indicators. The surveys in all countries utilized a standardized methodology and a two-stage stratified cluster sampling technique to select study participants. In the first stage, enumeration areas were randomly selected based on recent population data, and households were randomly selected in the second stage using the housing census as a sampling frame. For the final analysis, a total weighted sample of 101,983 women of reproductive age who had attended antenatal care during their index pregnancy was considered. Detailed description of DHS methodology is available on https://dhsprogram.com/Methodology/Survey-Types/DHS-Methodology.cfm.

### Variables and measurements

**Dependent variable.** In this study, the dependent variable was "late initiation of antenatal care visit". This variable was assessed using the timing of the first ANC visit during the most recent pregnancy. Women who commenced their first visit after three months of gestation were categorized as having late initiation, while those who initiated care earlier during the first three months of gestation were classified as having early antenatal visit [2, 28, 29].

**Independent variables.** Individual-level variables were grouped into sociodemographic and obstetric variables. Socio-demographic variables included current age, marital status, woman's and partner's education, women's employment status, media exposure, household head, wealth index, coverage by health insurance, and family size. Obstetric variables were age at marriage, number of under-5 children, birth order, birth interval, decision on health care, use of iron supplements, and history of pregnancy loss. While, residence, community-level media exposure, and community-level women illiteracy were the community-level variables.

Media exposure was computed using three variables (frequency of watching television, listening to the radio, and reading newspapers) that have three response options (i.e. not at all, less than once a week, and at least once a week). Thus, women who reported watching television, listening to the radio, or reading the newspaper at least once a week were considered as having media exposure and otherwise labeled as not having exposure to mass media.

Other community-level variables (i.e. community-level women illiteracy and media exposure) were generated by aggregating the individual-level observations at the cluster level and the aggregates were computed using the average values of the proportions of women in each

category of a given variable and median values were used to categorize the aggregated variables into two groups (i.e. low and high).

**Data management and statistical analysis.** Stata software version 17 was used for data cleaning and analysis. Before analysis, the availability of the outcome variable in the DHS dataset of each country was confirmed and all variables considered in the study were checked for missing values. Then, the datasets of 16 SSA countries were appended and a pooled weight was applied to compensate for the non-representativeness of the sample and obtain reliable estimates and standard errors.

To account for the clustering effects, a multilevel logistic regression analysis was applied to determine the effects of independent variables on the late initiation of ANC visit. Bivariable multilevel logistic regression analysis was done and all variables with a p-value of less than 0.25 in this analysis were considered for multivariable multilevel logistic regression analysis.

**Fixed-effect analysis.** In this analysis, we fitted four different models to determine the model that best fits the data: model I (a model with no independent variable to assess random variability in the intercept), model II (included only community-level variables), model III (included only individual-level variables), and model IV (included both individual and community-level variables). Then, log-likelihood (LL) and deviance (i.e. -2*LL) values were used for model selection and the model (i.e. model IV) with the lowest deviance and highest LL values was considered the best fit for the final analysis. The presence of multi-collinearity between explanatory variables was examined using generalized variance inflation factor values and this value was less than five for all variables, indicating no multi-collinearity was present. Finally, in the multivariable analysis, a p-value less than 0.05 and an adjusted odds ratio with the corresponding 95% confidence interval were used to identify the factors associated with late initiation of ANC visit.

**Random effect analysis.** Variations in the prevalence of late ANC visit among women across different clusters were assessed by intra-class correlation coefficient (ICC), proportion changes in variance (PCV), and median odds ratio (MOR). ICC quantifies the overall variance in the rate of late ANC visit that can be attributed to differences between clusters. ICC was calculated using the following formula:

$$\text{ICC} = \frac{V_a}{V_a + V_i}$$

Where, $V_a$ = cluster level variance and $V_i$ = individual level variance.

PCV (proportion of variance explained) assessed the differences in the prevalence of late initiation of ANC visit explained by community or individual-level characteristics in the model.

$$\text{PCV} = \frac{[V_a - V_b]}{V_a}$$

Where $V_a$ = the variance of the empty model and $V_b$ = the variance of the subsequent models.

The variation in the likelihood of late ANC visit at the cluster level was evaluated using the median odds ratio. This was determined by comparing the median probability of late ANC visits among women residing in areas with the highest risk of late visit to those living in clusters with the lowest risk. The MOR was estimated using the following formula:

$$\text{MOR} = \exp\{\sqrt{2\tau_0^2}\Phi(0.75)\}$$

Where Φ is the cumulative distribution function of a standard normal distribution and Φ* (0.75) ~ 0.6745.

### Ethical approval

Data used in this study were obtained from a DHS, and permission to access it was granted through an online request process available at http://www.dhsprogram.com. The survey procedures were also approved by the Institutional Review Board of the host country and ICF International. The accessed data were solely utilized for this registered study and can be publicly accessed from the program's official database.

## Results

### Socio-demographic characteristics

Of the total of 101,982 women, 46% of them were between the ages of 25–35 years, 84.9% were married, and more than one-third (34.9%) had never attended formal schooling. Furthermore, over three-fourths (79.6%) were from male-head households, about half (50.1%) had exposure to the media, and 60.9% were living in rural areas (Table 1).

### Obstetric and reproductive characteristics

Of the women included in the analysis, more than half (57.4%) were married at the age of eighteen or older, and about three-fourths (74.2%) had 1 to 2 children under the age of five years. Additionally, 30% of the participants had a preceding birth interval of less than 33 months (Table 2).

### Pooled prevalence of late ANC visit

The overall prevalence of late antenatal care visits among pregnant women in SSA countries was 53.8% [95% CI: 46.2% - 61.3%]. The prevalence varied across the countries, with the lowest level in Liberia (27.8%) and Mauritania (32.2%) and the highest in Nigeria (75.8%) followed by Kenya (72.2%) (Fig 1).

### Random effect result (measures of variations in late ANC visit)

The random effects analysis revealed that 19% of the variation in late initiation of antenatal visits was due to differences at the cluster level (as indicated by the ICC values from model I), while 16% of the variation (from model IV) was influenced by both individual and community-level factors. Additionally, the final model's PCV value suggests that the combined effect of these individual and community characteristics accounted for 71% of the variation in late ANC visits. Furthermore, the empty model's MOR of 2.07 indicates significant heterogeneity in late visit across different clusters. This means that women in clusters with a higher incidence of late visits were about two times more likely to attend ANC during late pregnancy compared to those in clusters with a lower rate of late visits (Table 3).

### Determinants of late initiation of antenatal care

The result of multilevel fixed-effect regression analysis showed that compared to older women, those in the age range of 15–24 years [AOR (95% CI) = 1.12 (1.05, 1.19)] had higher odds of attending antenatal care during late pregnancy. Similarly, the likelihood of late initiation of antenatal care was higher for women with primary education [AOR (95% CI) = 1.19 (1.13, 1.70)] when compared to those with higher education. Our analysis also revealed that women

**Table 1. Socio-demographic characteristics of women who attended antenatal care in SSA countries (n = 101,982).**

| Characteristics | Frequency (Weighted %) | Initiation of ANC visits | |
|---|---|---|---|
| | | Late | Early |
| **Age** | | | |
| 15–24 | 30,044 (29.5) | 17,864 (59.5) | 12,180 (40.5) |
| 25–34 | 46,920 (46.0) | 26,426 (56.3) | 20,494 (43.7) |
| 35–49 | 25,018 (24.5) | 14,479 (57.9) | 10,539 (42.1) |
| **Marital status** | | | |
| Never married | 8,574 (8.4) | 5,031 (58.7) | 3,543 (41.3) |
| Currently married | 86,628 (84.9) | 49,863 (57.6) | 36,767 (42.4) |
| Formerly married | 6,780 (6.7) | 3,876 (57.2) | 2,905 (42.8) |
| **Woman's education** | | | |
| No formal education | 35,592 (34.9) | 21,531 (60.5) | 14,061 (39.5) |
| Primary education | 30,116 (29.5) | 17,299 (57.4) | 12,818 (42.6) |
| Higher education | 36,274 (35.6) | 19,939 (55.0) | 16,335 (45.0) |
| **Partner education** | | | |
| No formal education | 30,210 (29.6) | 17,858 (59.1) | 12,351 (40.9) |
| Primary education | 22,309 (21.9) | 13,119 (58.8) | 9,190 (41.2) |
| Higher education | 49,463 (48.5) | 27,792 (56.2) | 21,671 (43.8) |
| **Woman's working status** | | | |
| Non-working | 36,866 (36.2) | 20,989 (56.9) | 15,877 (43.1) |
| Working | 65,116 (63.8) | 37,781 (58.0) | 27,336 (42.0) |
| **Media exposure** | | | |
| No | 50,970 (49.9) | 30,864 (60.6) | 20,106 (39.5) |
| Yes | 51,012 (50.1) | 27,905 (54.7) | 23,107 (45.3) |
| **Head of household** | | | |
| Male | 81,151 (79.6) | 47,491 (58.5) | 33,660 (41.5) |
| Female | 20,831 (20.4) | 11,278 (54.1) | 9,553 (45.9) |
| **Family size** | | | |
| 1–3 | 14,065 (13.8) | 7,708 (54.8) | 6,357 (45.2) |
| 4–6 | 43,930 (43.1) | 25,065 (57.1) | 18,866 (42.9) |
| > 6 | 43,987 (43.1) | 25,997 (59.1) | 17,990 (40.9) |
| **Household wealth** | | | |
| Poor | 40,273 (39.5) | 24,999 (62.1) | 15,274 (37.9) |
| Middle | 20,716 (20.3) | 12,537 (60.5) | 8,180 (39.5) |
| Rich | 40,993 (40.2) | 21,233 (51.8) | 19,760 (48.2) |
| **Covered by health insurance** | | | |
| No | 81,242 (91.2) | 47,959 (59.0) | 33,283 (41.0) |
| Yes | 7,883 (8.8) | 3,064 (38.9) | 4,819 (61.1) |
| **Residence** | | | |
| Urban | 39,892 (39.1) | 21,676 (54.3) | 18,217 (45.7) |
| Rural | 62,090 (60.9) | 37,094 (59.7) | 24,996 (40.3) |
| **Community-level women illiteracy** | | | |
| Low illiteracy | 52,986 (52.0) | 30,768 (58.1) | 22,218 (41.9) |
| High illiteracy | 48,996 (48.0) | 28,001 (57.2) | 20,995 (42.9) |
| **Community-level non-exposure to media** | | | |
| Low non-exposure | 53,269 (52.2) | 30,081 (56.5) | 23,188 (43.5) |
| High non-exposure | 48,713 (47.8) | 28,688 (58.9) | 20,025 (41.1) |

**Table 2. Obstetric and reproductive characteristics of women who attended antenatal care in SSA countries (n = 101,982).**

| Characteristics | Frequency (Weighted %) | Initiation of ANC visits | |
|---|---|---|---|
| | | Late | Early |
| **Age at marriage (n = 93,409)** | | | |
| < 18 year | 39,813 (42.6) | 24,528 (61.6) | 15,285 (38.4) |
| ≥ 18 year | 53,596 (57.4) | 29,210 (54.5) | 24,386 (45.5) |
| **Number of under-5 children** | | | |
| 0 | 4,841 (4.7) | 2,638 (54.5) | 2,203 (45.5) |
| 1–2 | 75,628 (74.2) | 42,966 (56.8) | 32,663 (43.2) |
| > 2 | 21,513 (21.1) | 13,165 (61.2) | 8,348 (38.8) |
| **Birth order** | | | |
| 1–3 | 60,660 (59.5) | 33,471 (55.2) | 27,189 (44.8) |
| > 3 | 41,322 (40.5) | 25,298 (61.2) | 16,025 (38.8) |
| **Birth interval** | | | |
| < 33 months | 30,599 (30.0) | 19,063 (62.3) | 11,536 (37.7) |
| ≥ 33 months | 71,383 (70.0) | 39,706 (55.6) | 31,678 (44.4) |
| **Decision on healthcare** | | | |
| Women alone | 15,236 (14.9) | 8,944 (58.7) | 6,292 (41.3) |
| Jointly with husband | 33,470 (32.8) | 18,397 (55.0) | 15,072 (45.0) |
| Others* | 53,276 (52.2) | 31,428 (59.0) | 21,849 (41.0) |
| **Number of ANC visits** | | | |
| < 8 visits | 91,933 (90.2) | 55,349 (60.2) | 36,585 (39.8) |
| ≥ 8 visits | 10,049 (9.9) | 3,420 (34.0) | 6,629 (66.0) |
| **Given iron supplement during ANC visit** | | | |
| No | 8,703 (8.5) | 5,649 (64.9) | 3,055 (35.1) |
| Yes | 93,279 (91.5) | 53,121 (57.0) | 40,159 (43.0) |
| **Days iron supplement consumed (n = 86,859)** | | | |
| < 90 days | 42,858 (49.3) | 26,840 (62.6) | 16,018 (37.4) |
| ≥ 90 days | 44,000 (50.7) | 23,045 (52.4) | 20,956 (47.6) |
| **Place of delivery** | | | |
| Home | 25,163 (24.7) | 17,838 (70.9) | 7,325 (29.1) |
| Health facility | 76,819 (75.3) | 40,931 (53.3) | 35,888 (46.7) |
| **Ever terminated pregnancy** | | | |
| No | 87,303 (85.6) | 50,942 (58.4) | 36,361 (41.7) |
| Yes | 14,680 (14.4) | 7,827 (53.3) | 6,853 (46.7) |

Others* = husband/partner alone, someone else, and other DHS category.

enrolled in the health insurance scheme [AOR (95% CI) = 2.05 (1.90, 2.21)] and those with higher birth order [AOR (95% CI) = 1.28 (1.22, 1.34)] had higher odds of late antenatal care visit than women not enrolled in this program and those with the birth order less than four, respectively. Additionally, partner's education, household wealth, age at marriage, women's working status, decision on health care, residence, and community-level women's illiteracy were the other factors associated with late initiation of antenatal care (Table 4).

## Discussion

This analysis aimed to assess the prevalence of late antenatal care visit and its determinants among pregnant women in SSA. The study found that 53.8% [95% CI: 46.2% - 61.3%] of women had late antenatal care visit, a rate consistent with that reported from the studies done

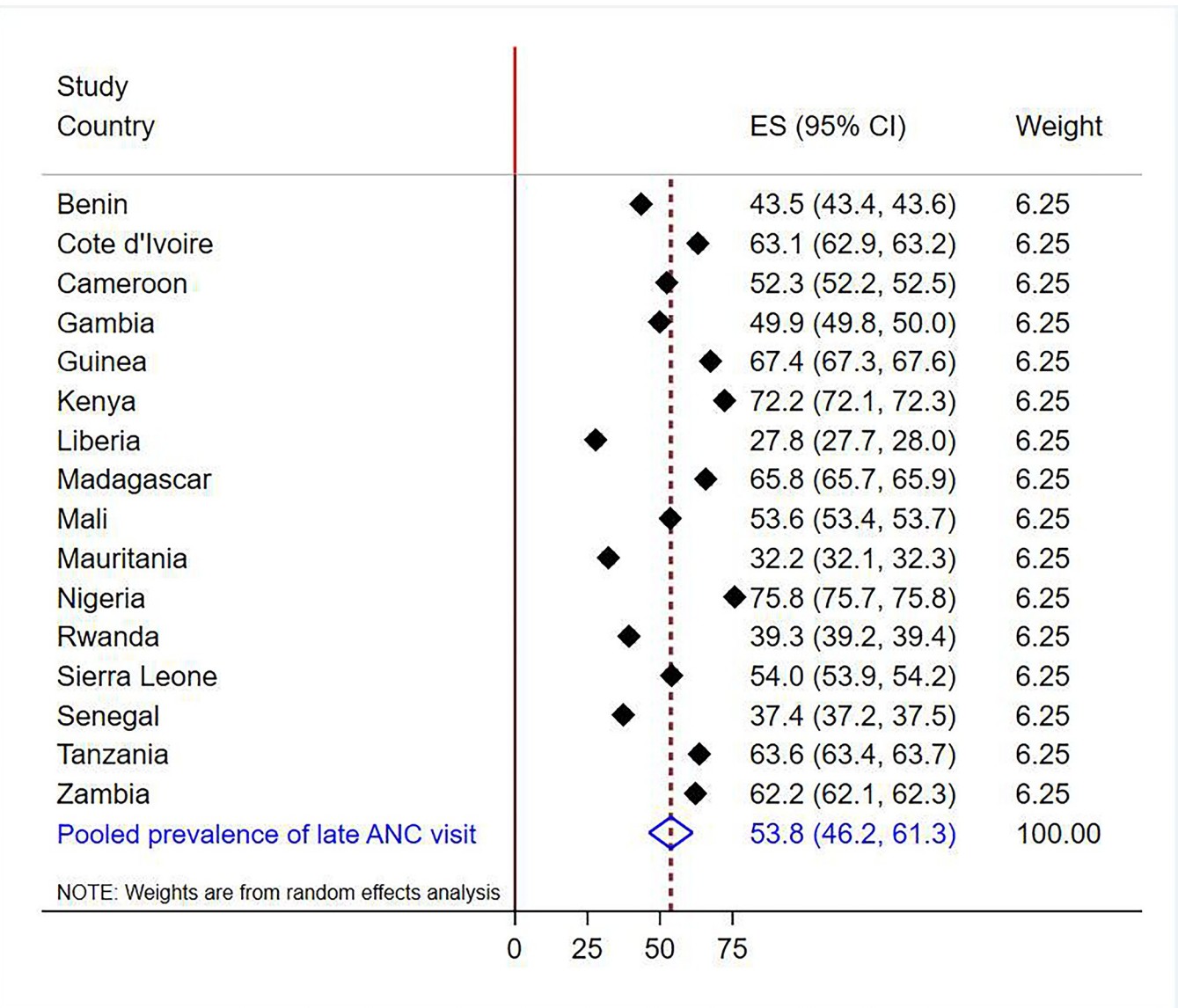

**Fig 1. Pooled and country-level prevalence of late antenatal care visits among pregnant women in SSA, 2018–2022.**

**Table 3. Measures of variations and model comparison for determinants of late antenatal care visit among pregnant women in SAA countries.**

| Measure of variation | Model 1 | Model 2 | Model 3 | Model 4 |
|---|---|---|---|---|
| Cluster-level variance (%) | 64 | 20 | 41 | 18 |
| Intra-class correlation (%) | 19 | 13 | 18 | 16 |
| Proportion change in variance (%) | Reference | 69 | 36 | 71 |
| Median odds ratio | 2.07 | 1.16 | 1.66 | 1.10 |
| **Model comparison** | | | | |
| Akaike's information criteria | 137,196 | 108,119 | 136,886 | 108,117 |
| Bayesian information criteria | 137,215 | 108,296 | 136,934 | 108,266 |
| Log-likelihood | -68,596 | -54,043 | -68,438 | -54,040 |
| Deviance | 137,192 | 108,086 | 136,876 | 108,080 |

**Table 4. A multilevel fixed-effects analysis of determinants of late antenatal care visits among pregnant women in SSA countries.**

| Individual-level determinants | Model II | Model III | Model IV |
|---|---|---|---|
| **Age** | | | |
| 15–24 | 1.12 (1.05, 1.18) | - | 1.12 (1.05, 1.19)* |
| 25–34 | 1.02 (0.97, 1.06) | | 1.02 (0.97, 1.06) |
| 35–49 | 1.00 | | 1.00 |
| **Woman's education** | | | |
| No formal education | 1.20 (1.14, 1.30) | - | 1.19 (1.13, 1.70)* |
| Primary education | 1.00 (0.93, 1.05) | | 1.03 (0.97, 1.08) |
| Higher education | 1.00 | | 1.00 |
| **Partner education** | | | |
| No formal education | 1.03 (0.97, 1.09) | - | 1.00 (0.95, 1.04) |
| Primary education | 1.11 (1.05, 1.17) | | 1.15 (1.10, 1.22)* |
| Higher education | 1.00 | | 1.00 |
| **Women's working status** | | | |
| Non-working | 1.00 | | 1.00 |
| Working | 1.12 (1.07, 1.17) | - | 1.12 (1.07, 1.17)* |
| **Household wealth** | | | |
| Poor | 1.36 (1.28, 1.44) | - | 1.36 (1.27, 1.45)* |
| Middle | 1.34 (1.26, 1.41) | | 1.33 (1.25, 1.42)* |
| Rich | 1.00 | | 1.00 |
| **Age at marriage** | | | |
| < 18 year | 1.13 (1.09, 1.18) | - | 1.13 (1.08, 1.17)* |
| > 18 year | 1.00 | | 1.00 |
| **Birth order** | | | |
| 1–3 | 1.00 | | 1.00 |
| > 3 | 1.28 (1.22, 1.33) | - | 1.28 (1.22, 1.34)* |
| **Covered by health insurance** | | | |
| Yes | 1.00 | | 1.00 |
| No | 2.06 (1.91, 2.22) | - | 2.05 (1.90, 2.21)* |
| **Decision on healthcare** | | | |
| Jointly with husband | 1.00 | | 1.00 |
| Women alone | 1.11 (1.05, 1.18) | - | 1.11 (1.05, 1.18)* |
| Others* | 1.16 (1.11, 1.21) | | 1.16 (1.11, 1,22)* |
| **Community-level determinants** | | | |
| **Residence** | | | |
| Urban | | 1.00 | 1.00 |
| Rural | - | 1.29 (1.22, 1.36) | 1.25 (1.16, 1.28)* |
| **Community-level women illiteracy** | | | |
| Low illiteracy | | 1.00 | 1.00 |
| High illiteracy | - | 1.11 (1.04, 1.18) | 1.22 (1.14, 1.30)* |
| **Community-level non-exposure to media** | | | |
| Low non-exposure | - | 0.95 (0.90, 1.02) | 0.94 (0.88, 1.01) |
| High non-exposure | | 1.00 | 1.00 |

*Statistically significant at p-value less than 0.05.

in central (47%) [19] and Southwestern (48%) [21] Ethiopia and Rwanda (61.1%) [17]. However, this finding is lower compared to the results of the studies in Southern Ethiopia (66%) [30], Uganda (73%) [16], and Tanzania (85.3%) [14], but higher than the result of the studies in Ethiopia (32%) [11] and Afghanistan (44.2%) [29]. Potential explanations for this discrepancy may include the use of different definitions for late ANC visit. For example, a study conducted in Ethiopia [11] defined late ANC as a first visit occurring after four months of gestation, in contrast to the three-month threshold used in this study. Additionally, differences in sociodemographic characteristics of participants across the studies may also play a role in the observed disparities.

Our analysis showed that women aged 15–24 years were more likely to initiate antenatal care during late pregnancy when compared to older women. This finding is inconsistent with the previous studies in Ethiopia [12, 19, 30] and the Democratic Republic of Congo [15] which reported higher odds of late visits among older women. The most possible justification for the observed inconsistencies would be due to differences in the study setting, as most of the previous studies [12, 19, 30] were institution-based, whereas the current study used community-based cross-sectional data. Additionally, the association between younger maternal age and late antenatal visit can be explained by the fact that younger women may have less experience with pregnancy and childbirth, leading them to delay seeking care during the later trimesters, as they may not recognize the importance of early care.

According to this study, women with primary education were 1.19 times more likely to have late antenatal care visits than those with higher education. This finding is supported by the previous studies done in Bangladesh [18], the Democratic Republic of Congo [15], Ethiopia [12, 30], Eastern African countries [13], Nigeria [4], and Tanzania [14]. This finding can be explained by the fact that women with lower educational status may have limited access to information about the importance of early antenatal care, leading to delays in seeking care. Moreover, lower education attainment may be associated with lower socioeconomic status, making it difficult for women to prioritize antenatal care appointments over other competing priorities.

Our analysis showed higher odds of late antenatal visits among women whose partners had no formal and primary-level education compared to those whose husbands had higher educational attainment. This finding is in line with the results of the studies in Southern Ethiopia and Bangladesh [18, 30]. This might be because limited health literacy among men with lower education may contribute to a lack of understanding about the significance of early antenatal visits, leading to delays in seeking care for their spouses.

This study revealed that women who did not have health insurance coverage were more likely to book antenatal care during late pregnancy compared to those enrolled in the health insurance scheme. This result is in agreement with the findings of the studies in South Ethiopia [30] and Rwanda [17]. The possible reason could because women without health insurance coverage may have limited access to health care services due to financial constraints that hinder them from seeking care early in their pregnancies, resulting in visit appointments later in their pregnancies.

Household wealth was also found to have a significant association with late ANC visits. For instance, the odds of late antenatal visits were higher among women from poor and middle-class households compared to those from rich households. This finding is supported by the result of the previous studies [11, 13, 15], which reported an inverse relationship between household wealth and the timing of prenatal care. This might be because women from households with low socio-economic status may have limited financial resources, making it more difficult for them to prioritize antenatal care appointments and seek healthcare services early in their pregnancies.

In this study, women of higher birth order had a 28% higher likelihood of late antenatal care visits compared to women belonging to low birth order. This finding is in agreement with studies in Ethiopia [11], Eastern Africa, and Rwanda [17]. This might be because women with higher birth orders may have increased responsibilities within their families, which can impact their ability to prioritize their own healthcare needs, including antenatal care. Additionally, socioeconomic factors associated with higher birth order, such as limited financial resources or lack of social support, may contribute to delays in visit for antenatal care.

Our result also showed that women married during their early age had higher odds of late ANC visit compared to those married at the age of 18 or beyond. One possible reason for this finding could be that women who marry at a younger age may face additional social, economic, or cultural pressures that make it more difficult for them to prioritize their health and seek timely care [31]. Moreover, these women may have limited autonomy and decision-making power within their marriages, leading to delays in seeking healthcare services. Additionally, early marriage may be associated with lower education and awareness about the importance of ANC, further contributing to late visits [32].

Consistent with the findings of the studies in Ethiopia [11], and East Africa [13, 18], this study found a significant relationship between the place of residence and timing of antenatal care visits, in which there were higher odds of delayed visits among women from rural residences. Limited access to healthcare facilities in rural areas might have resulted in delayed antenatal visits among rural women, as women may have to travel long distances to reach a healthcare facility. Additionally, socioeconomic factors in rural areas, such as lower income levels and limited education, may contribute to delayed antenatal visits as women may prioritize other expenses over seeking healthcare.

## Strengths and limitations

The study's primary strength lies in its utilization of data from a nationally representative DHS conducted in 16 Sub-Saharan African countries. Data were collected using a validated tool and involved a large sample size. Additionally, the study employed advanced statistical techniques, i.e. multilevel modeling, which accounted for the hierarchical structure of the DHS data. However, it is important to note that due to the cross-sectional nature of the data, it is not possible to establish a causal relationship between the explanatory and outcome variables. Furthermore, there may be recall bias as participants were asked to recall events that occurred five years or more before the survey.

## Conclusion

This study showed that more than half of pregnant women in SSA had made their first ANC visits during late pregnancy, with large variations observed across the countries. Younger maternal age, lower education levels, lack of health insurance, higher birth order, partner's education, household wealth, age at marriage, working status, healthcare decision-making power, residence, and community-level women's illiteracy were associated with late initiation of antenatal care. Thus, health education and awareness creation programs targeting younger women, those with lower education, and women from families of low socioeconomic status are important to improve early antenatal care visits. Moreover, strengthening the existing interventions focusing on increasing women's involvement in health care decisions, improving access to health care in rural settings, preventing early marriage, and promoting health insurance enrollment is also crucial.

## Supporting information

**S1 Dataset. Dataset for the study on late initiation of antenatal care visit amid implementation of new antenatal care model in Sub-Saharan African countries.**
(XLSX)

## Author Contributions

**Conceptualization:** Kusse Urmale Mare, Gashaye Gobena Andargie, Abdulkerim Hassen Moloro, Beriso Furo Wengoro, Begetayinoral Kussia Lahole, Tesfahun Simon Hadaro, Simeon Meskele Leyto, Petros Orkaido Mamo, Temesgen Gebeyehu Wondmeneh, Oumer Abdulkadir Ebrahim, Kebede Gemeda Sabo.

**Data curation:** Kusse Urmale Mare, Begetayinoral Kussia Lahole, Simeon Meskele Leyto, Beminate Lemma Seifu, Temesgen Gebeyehu Wondmeneh, Kebede Gemeda Sabo.

**Formal analysis:** Kusse Urmale Mare, Gashaye Gobena Andargie, Abdulkerim Hassen Moloro, Ahmed Adem Mohammed, Osman Ahmed Mohammed, Beriso Furo Wengoro, Begetayinoral Kussia Lahole, Simeon Meskele Leyto, Petros Orkaido Mamo, Abdulhakim Hora Hedato, Oumer Abdulkadir Ebrahim, Kebede Gemeda Sabo.

**Investigation:** Kusse Urmale Mare.

**Methodology:** Kusse Urmale Mare, Gashaye Gobena Andargie, Ahmed Adem Mohammed, Osman Ahmed Mohammed, Beriso Furo Wengoro, Begetayinoral Kussia Lahole, Tesfahun Simon Hadaro, Simeon Meskele Leyto, Petros Orkaido Mamo, Abdulhakim Hora Hedato, Beminate Lemma Seifu, Temesgen Gebeyehu Wondmeneh, Oumer Abdulkadir Ebrahim, Kebede Gemeda Sabo.

**Resources:** Kusse Urmale Mare.

**Software:** Kusse Urmale Mare, Gashaye Gobena Andargie, Abdulkerim Hassen Moloro, Ahmed Adem Mohammed, Osman Ahmed Mohammed, Beriso Furo Wengoro, Begetayinoral Kussia Lahole, Tesfahun Simon Hadaro, Simeon Meskele Leyto, Abdulhakim Hora Hedato, Beminate Lemma Seifu, Temesgen Gebeyehu Wondmeneh, Oumer Abdulkadir Ebrahim, Kebede Gemeda Sabo.

**Validation:** Kusse Urmale Mare.

**Visualization:** Kusse Urmale Mare, Simeon Meskele Leyto.

**Writing – original draft:** Kusse Urmale Mare, Gashaye Gobena Andargie, Abdulkerim Hassen Moloro, Osman Ahmed Mohammed, Begetayinoral Kussia Lahole, Tesfahun Simon Hadaro, Simeon Meskele Leyto, Petros Orkaido Mamo, Abdulhakim Hora Hedato, Temesgen Gebeyehu Wondmeneh, Oumer Abdulkadir Ebrahim, Kebede Gemeda Sabo.

**Writing – review & editing:** Gashaye Gobena Andargie, Abdulkerim Hassen Moloro, Ahmed Adem Mohammed, Osman Ahmed Mohammed, Beriso Furo Wengoro, Begetayinoral Kussia Lahole, Tesfahun Simon Hadaro, Petros Orkaido Mamo, Abdulhakim Hora Hedato, Beminate Lemma Seifu, Temesgen Gebeyehu Wondmeneh, Oumer Abdulkadir Ebrahim, Kebede Gemeda Sabo.

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
