## [Decision Letter · Decision Letter 0]

10 Jul 2024

PONE-D-24-21779Late initiation of antenatal care visit amid implementation of new antenatal care model in Sub-Saharan African countries: a multilevel analysis of multination population survey data PLOS ONE

Dear Dr. Mare,

Thank you for submitting your manuscript to PLOS ONE. After careful consideration, we feel that it has merit but does not fully meet PLOS ONE’s publication criteria as it currently stands. Therefore, we invite you to submit a revised version of the manuscript that addresses the points raised during the review process.

We look forward to receiving your revised manuscript.

Kind regards,

Hassen Mosa, Msc

Academic Editor

PLOS ONE

Journal Requirements:

Reviewers' comments:

Reviewer's Responses to Questions

**Comments to the Author**

1. Is the manuscript technically sound, and do the data support the conclusions?

Reviewer #1: Yes

Reviewer #2: Yes

2. Has the statistical analysis been performed appropriately and rigorously? 

Reviewer #1: Yes

Reviewer #2: Yes

3. Have the authors made all data underlying the findings in their manuscript fully available?

Reviewer #1: No

Reviewer #2: Yes

4. Is the manuscript presented in an intelligible fashion and written in standard English?

Reviewer #1: Yes

Reviewer #2: Yes

5. Review Comments to the Author

**Reviewer #1:** • Line 78-80: “Furthermore, comprehensive health system and community-level interventions have been implemented to bolster the quality and accessibility of ANC services and community awareness about the importance of timely ANC initiation, respectively” The authors should provide citations and references to back this statement

• Results: I will suggest the author report the proportion or the percentages only without quoting the numbers of participants

• Where are the tables the authors are referring to in their results?

• Tables should be placed directly after the next reporting result from the table

• Table 1 should be quoted in the text where the results are reported

• Discussion: Though the authors make a comparison between their findings and those of previous studies, it will be important they provide the public health implication of their findings

• The tables on multilevel analysis results provided by the authors are difficult to understand, I will suggest the authors make their tables as concise as possible showing all the three models of the multilevel regression carried out

**Reviewer #2:** This is an insightful study that addresses the hurdle of maternal health, which is late initiation of ANC by using the latest model. The manuscript is readable, however the conclusion is too general and I would suggest authors get it edited.

6. PLOS authors have the option to publish the peer review history of their article (what does this mean?). If published, this will include your full peer review and any attached files.

Reviewer #1: **Yes: **Robert Kokou Dowou

Reviewer #2: No

---

## [Author Response · Author response to Decision Letter 0]

13 Jul 2024

Thank you for the opportunity to revise our manuscript “Late initiation of antenatal care visit amid implementation of new antenatal care model in Sub-Saharan African countries: a multilevel analysis of multination population survey data (ID: PONE-D-24-21779)”. We have addressed the concerns raised by the reviewers using a point-by-point response as stated below. The amendments made to the manuscript have been presented using track change in the attachment titled “Revised Manuscript with Track Changes”.

Journal Requirements

Comment 1: When submitting your revision, we need you to address these additional requirements. Please ensure that your manuscript meets PLOS ONE's style requirements, including those for file naming. The PLOS ONE style templates can be found at https://journals.plos.org/plosone/s/file?id=wjVg/PLOSOne_formatting_sample_main_body.pdf and https://journals.plos.org/plosone/s/file?id=ba62/PLOSOne_formatting_sample_title_authors_affiliations.pdf.

Response 1: Thank you. We have amended the manuscript as per the PLOS ONE style requirements.

Comment 2: Thank you for uploading your study's underlying data set. Unfortunately, the repository you have noted in your Data Availability statement does not qualify as an acceptable data repository according to PLOS's standards. At this time, please upload the minimal data set necessary to replicate your study's findings to a stable, public repository (such as Figshare or Dryad) and provide us with the relevant URLs, DOIs, or accession numbers that may be used to access these data. For a list of recommended repositories and additional information on PLOS standards for data deposition, please see https://journals.plos.org/plosone/s/recommended-repositories.

Response 2: Thank you very much for your recommendation. We have provided the relevant URL for the data used in the study and uploaded the minimal dataset used for the analysis.

Comment 3: Please review your reference list to ensure that it is complete and correct. If you have cited papers that have been retracted, please include the rationale for doing so in the manuscript text, or remove these references and replace them with relevant current references. Any changes to the reference list should be mentioned in the rebuttal letter that accompanies your revised manuscript. If you need to cite a retracted article, indicate the article’s retracted status in the References list and also include a citation and full reference for the retraction notice.

Response 3: Thank you very much. We have checked the reference list and ensured that the list does not contain a retracted paper.

Responses to Reviewer #1 Comments

Comment 1: Line 78-80: “Furthermore, comprehensive health system and community-level interventions have been implemented to bolster the quality and accessibility of ANC services and community awareness about the importance of timely ANC initiation, respectively” The authors should provide citations and references to back this statement.

Response 1: Thank you very much for your important recommendation. We have provided citations and references for the stated statement as suggested. 

Comment 2: Results: I will suggest the author report the proportion or the percentages only without quoting the number of participants.

Response 2: Thank you for your suggestion. We have incorporated this suggestion into the descriptive results of the revised manuscript. 

Comment 3: Where are the tables the authors are referring to in their results? Tables should be placed directly after the next reporting result from the table. Table 1 should be quoted in the text where the results are reported.

Response 3: Thank you for your concern. We agree that tables should be placed right after their respective descriptive results, however, as per the journal requirements, tables are placed at the end of the main body of the manuscript just following the reference list. Thus, details of the tables quoted in the main body are available at the end of the manuscript. 

Comment 4: Discussion-though the authors make a comparison between their findings and those of previous studies, it will be important they provide the public health implication of their findings.

Response 4: Thank you very much. We have provided the public health implication of the study findings under the conclusion section of the revised manuscript.

Comment 5: The tables on multilevel analysis results provided by the authors are difficult to understand, I will suggest the authors make their tables as concise as possible showing all the three models of the multilevel regression carried out.

Response 5: Thank you very much for your important recommendation. We have presented the results of multilevel regression analysis for the three models as suggested in the revised manuscript.

Responses to Reviewer #2 Comments

This is an insightful study that addresses the hurdle of maternal health, which is late initiation of ANC by using the latest model. 

Comment: The manuscript is readable, however, the conclusion is too general and I would suggest authors get it edited.

Response: Thank you for your interesting suggestion. We have revised the conclusion section as suggested.

---

## [Decision Letter · Decision Letter 1]

16 Oct 2024

PONE-D-24-21779R1Late initiation of antenatal care visit amid implementation of new antenatal care model in Sub-Saharan African countries: a multilevel analysis of multination population survey dataPLOS ONE

Dear Dr. Mare,

Thank you for submitting your manuscript to PLOS ONE. After careful consideration, we feel that it has merit but does not fully meet PLOS ONE’s publication criteria as it currently stands. Therefore, we invite you to submit a revised version of the manuscript that addresses the points raised during the review process.

To ensure a thorough evaluation of your methodological approach, an additional reviewer was consulted in this round. The reviewer has expressed a number of major concerns about the quality of the reporting and analysis, particularly with respect to your statistical approach. Please ensure you carefully revise the manuscript to address all of the concerns raised.

We look forward to receiving your revised manuscript.

Kind regards,

Marianne Clemence

Staff Editor

PLOS ONE

Reviewers' comments:

Reviewer's Responses to Questions

**Comments to the Author**

1. If the authors have adequately addressed your comments raised in a previous round of review and you feel that this manuscript is now acceptable for publication, you may indicate that here to bypass the “Comments to the Author” section, enter your conflict of interest statement in the “Confidential to Editor” section, and submit your "Accept" recommendation.

Reviewer #1: All comments have been addressed

Reviewer #2: All comments have been addressed

Reviewer #3: (No Response)

2. Is the manuscript technically sound, and do the data support the conclusions?

Reviewer #1: Yes

Reviewer #2: Yes

Reviewer #3: Partly

3. Has the statistical analysis been performed appropriately and rigorously? 

Reviewer #1: Yes

Reviewer #2: Yes

Reviewer #3: No

4. Have the authors made all data underlying the findings in their manuscript fully available?

Reviewer #1: Yes

Reviewer #2: Yes

Reviewer #3: Yes

5. Is the manuscript presented in an intelligible fashion and written in standard English?

Reviewer #1: Yes

Reviewer #2: Yes

Reviewer #3: Yes

6. Review Comments to the Author

Reviewer #1: Data source: The authors should provide the accession number or the direct web link in the Availability of Data and Materials section of your manuscript.

Reviewer #2: The authors have addressed all the comments given in the revision one of the manuscript. I have no further comment.

Reviewer #3: The paper addresses an important topic. However, it has some major technical weaknesses. In particular, some crucial details on methodology are missing, making it impossible to assess appropriateness of the analysis presented.

1. It is stated that the analysis is based on multilevel analysis, yet it is unclear what the different levels are. For instance, is the multilevel analysis based on two-levels or 3-level logistic regression, and what are the levels (community, country, etc)? The results presented only show fixed effects without any random effects, so it is unclear whether there were any significant variations across the different levels of analysis. It is impossible to ascertain what multilevel models were fitted, or results based on single-level logistic regression.

2. It is stated that weighting was undertaken for pooled data, but unclear how this was done. Was weighting based of DHS-derived country-specific weights, or were pooled weights derived, taking into consideration different sampling fractions used by DHS in different countries? The DHS-derived weights are useful for country-specific analysis to achieve national representativeness, but would not be appropriate for pooled data across different countries in sub-Saharan Africa.

Other points for consideration:

- Need to correct errors in some factual statements – e,g. first paragraph of introduction ‘…the services received after first trimester are considered late’ . Do you mean to say initiation/booking of ANC after first semester is considered late?

- It is suggested that ‘…evidence on the level of late antenatal care booking and its determinants in SSA following the implementation of the 2016 new antenatal care model is limited’, but it is unclear how the new ANC model deviates from previous guidelines in relation to initiation/timing of ANC.

- Table 6 presents ‘multilevel fixed-effects’ analysis –significance for some of the factors where ORs suggest significance are not marked so, e.g rural residence and decision on healthcare. Are these omissions or intended? Clarification needed if the latter.

- Reference to table 3 in results section, yet there is no Table 3 among the list of tables provided at the end of the paper. Is this the same as table 6?

- Aspects of discussion could be better thought through. For instance, the lower prevalence of late ANC compared to other studies in this paper has been attributed to differences in sample size and scope of study, the current study having a much larger sample covering a wider geographic area. It is important to note that a larger sample size in itself is unlikely to lead to lower prevalence. Potential explanations may include use of different definition for late ANC – other studies (e.g. [14] defined late ANC as 2nd or 3rd trimester (after 1st trimester – i.e 3 months) rather than after 4 months used in the study.

- Reporting of results could be better thought through. Eg it is stated that the analysis ‘revealed that health insurance coverage [AOR (95% CI) = 1.84 (1.68, 2.01)] and higher birth order 184 [AOR (95% CI) = 1.40 (1.34, 1.47)] were associated with higher odds of late antenatal care booking’ without clarifying what is being compared or what the Odds ratios refer to.

7. PLOS authors have the option to publish the peer review history of their article (what does this mean?). If published, this will include your full peer review and any attached files.

Reviewer #1: **Yes: **Robert Kokou Dowou

Reviewer #2: No

Reviewer #3: No

---

## [Author Response · Author response to Decision Letter 1]

11 Nov 2024

Thank you for the opportunity to revise our manuscript. “Late initiation of antenatal care visit amid implementation of new antenatal care model in Sub-Saharan African countries: a multilevel analysis of multination population survey data (ID: PONE-D-24-21779R1)”. We have addressed the concerns raised by the reviewers using a point-by-point response as stated below. The amendments made to the manuscript have been presented using track change in the attachment titled “Revised Manuscript with Track Changes”.

Response to Reviewer #1 Comment

Comment 1: Data source: The authors should provide the accession number or the direct web link in the Availability of Data and Materials section of your manuscript.

Response 1: Thank you very much for your suggestion. We have provided the web link and a reference number of an authorization latter to access the data used in our analysis.

Response to Reviewer #2 Comment

Comment: The authors have addressed all the comments given in the revision one of the manuscript. I have no further comment.

Response: Thank you very much for accepting our manuscript in its current form.

Response to Reviewer #3 Comments:

The paper addresses an important topic. However, it has some major technical weaknesses. In particular, some crucial details on methodology are missing, making it impossible to assess appropriateness of the analysis presented.

Comment 1: It is stated that the analysis is based on multilevel analysis, yet it is unclear what the different levels are. For instance, is the multilevel analysis based on two-levels or 3-level logistic regression, and what are the levels (community, country, etc)? The results presented only show fixed effects without any random effects, so it is unclear whether there were any significant variations across the different levels of analysis. It is impossible to ascertain what multilevel models were fitted, or results based on single-level logistic regression.

Response 1: Thank you very much for your important concern. We have presented the random effect analysis result in the revised manuscript (i.e. we have revised the method and result section based on the suggestion). 

Comment 2: It is stated that weighting was undertaken for pooled data, but unclear how this was done. Was weighting based of DHS-derived country-specific weights, or were pooled weights derived, taking into consideration different sampling fractions used by DHS in different countries? The DHS-derived weights are useful for country-specific analysis to achieve national representativeness, but would not be appropriate for pooled data across different countries in sub-Saharan Africa.

Response 2: Thank you again for your interesting comment. We have generated a pooled weight and used it throughout our analysis and we have indicated this issue as per your recommendation in the revised manuscript. 

Other points for consideration:

Comment 3: Need to correct errors in some factual statements – e,g. first paragraph of introduction ‘…the services received after first trimester are considered late’ . Do you mean to say initiation/booking of ANC after first semester is considered late?

Response 3: Thank you and sorry for the grammatical error. We have replaced it with the suggested phrase. 

Comment 4: It is suggested that ‘…evidence on the level of late antenatal care booking and its determinants in SSA following the implementation of the 2016 new antenatal care model is limited’, but it is unclear how the new ANC model deviates from previous guidelines in relation to initiation/timing of ANC.

Response 4: Thank you for your important comment. As stated in paragraph 5 of an “Introduction” section, the new recommendation deviates from the former one by the minimum number of ANC visits that a pregnant woman should attend during pregnancy. For instance, while the former model recommends a minimum of four visits, the new guideline suggests a minimum of eight visits. 

Comment 5: Table 6 presents ‘multilevel fixed-effects’ analysis –significance for some of the factors where ORs suggest significance are not marked so, e.g rural residence and decision on healthcare. Are these omissions or intended? Clarification needed if the latter.

Response 5: Thank you very much for your important comment. It is an editorial error and we have marked them in the revision.

Comment 6: Reference to table 3 in results section, yet there is no Table 3 among the list of tables provided at the end of the paper. Is this the same as table 6?

Response 6: Thank you for your concern. Yes, it is the same as table 6 and we have corrected this issue in the revised manuscript. 

Comment 7: Aspects of discussion could be better thought through. For instance, the lower prevalence of late ANC compared to other studies in this paper has been attributed to differences in sample size and scope of study, the current study having a much larger sample covering a wider geographic area. It is important to note that a larger sample size in itself is unlikely to lead to lower prevalence. Potential explanations may include use of different definition for late ANC – other studies (e.g. [14] defined late ANC as 2nd or 3rd trimester (after 1st trimester – i.e. 3 months) rather than after 4 months used in the study.

Response 7: Thank you for your important recommendation. We have considered the suggested explanation/justification for the observed discrepancy in the revision.

Comment 8: Reporting of results could be better thought through. Eg it is stated that the analysis ‘revealed that health insurance coverage [AOR (95% CI) = 1.84 (1.68, 2.01)] and higher birth order 184 [AOR (95% CI) = 1.40 (1.34, 1.47)] were associated with higher odds of late antenatal care booking’ without clarifying what is being compared or what the Odds ratios refer to.

Response 8: Thank you very much. We have revised this concern and provided the reference category (comparison group) for the stated factors.

---

## [Decision Letter · Decision Letter 2]

5 Dec 2024

PONE-D-24-21779R2Late initiation of antenatal care visit amid implementation of new antenatal care model in Sub-Saharan African countries: a multilevel analysis of multination population survey dataPLOS ONE

Dear Dr. Mare,

Thank you for submitting your manuscript to PLOS ONE. After careful consideration, we feel that it has merit but does not fully meet PLOS ONE’s publication criteria as it currently stands. Therefore, we invite you to submit a revised version of the manuscript that addresses the points raised during the review process.

Thank you for addressing the initial comments provided by the reviewers. Based on the revised manuscript, the reviewers and I have requested that some minor comments still need to be addressed.

We look forward to receiving your revised manuscript.

Kind regards,

Muhammad Haroon Stanikzai

Academic Editor

PLOS ONE

Journal Requirements:

Additional Editor Comments:

- Definition of the outcome – WHO recommends 1st ANC visit within 3 months (1st trimester). In the manuscript authors defined late ANC after 4 months of gestation. According to the literature, ANC visit within 4 months is also consider late ANC. Additionally, authors compared their results with papers mostly discussing late ANC after 3 months. I ask the authors to revisit WHO recommendations for ANC timing and revised their results accordingly. I also suggest the authors can consult and add the following citations which details late ANC in another low-income country.

--https://journals.plos.org/plosone/article?id=10.1371/journal.pone.0309300

--https://www.dovepress.com/sociodemographic-predictors-of-initiating-antenatal-care-visits-by-pre-peer-reviewed-fulltext-article-IJWH

- In Abstract add the abbreviation SSA from the first paragraph (line 25).

- In Abstract add the full form (DHS) from the first paragraph (line 27).

- Methods described aren't the ones opted by the author. They were done part of the National Surveys. These methods should be supported be relevant citations (lines 94-100).

- Dependent variable: Add citation considering my earlier comment.

Reviewers' comments:

Reviewer's Responses to Questions

**Comments to the Author**

1. If the authors have adequately addressed your comments raised in a previous round of review and you feel that this manuscript is now acceptable for publication, you may indicate that here to bypass the “Comments to the Author” section, enter your conflict of interest statement in the “Confidential to Editor” section, and submit your "Accept" recommendation.

Reviewer #2: All comments have been addressed

Reviewer #4: All comments have been addressed

2. Is the manuscript technically sound, and do the data support the conclusions?

Reviewer #2: Yes

Reviewer #4: Yes

3. Has the statistical analysis been performed appropriately and rigorously? 

Reviewer #2: Yes

Reviewer #4: Yes

4. Have the authors made all data underlying the findings in their manuscript fully available?

Reviewer #2: Yes

Reviewer #4: Yes

5. Is the manuscript presented in an intelligible fashion and written in standard English?

Reviewer #2: Yes

Reviewer #4: Yes

6. Review Comments to the Author

Reviewer #2: The authors addressed all comments given in the revised version of the manuscript. Therefore, I have no further comments.

Reviewer #4: Dear Editors and Authors,

Thank you for giving me the opportunity to review the manuscript, titled “Late initiation of antenatal care visit amid implementation of new antenatal care model in Sub-Saharan African countries: a multilevel analysis of multination population survey data”.

The authors of the study chose a very important topic related to maternal health in Africa. This study aimed to determine the pooled prevalence of late ANC visit and its determinants among women in Sub-Saharan Africa (SSA) using DHS conducted after the implementation of new guidelines in 2016.

The authors used from 16 SSA between 2018 and 2022, with a weighted sample of 101,983 women who had ANC visits during their pregnancy. The authors applied logistic regression models for multivariate results and found the pooled prevalence of late ANC visits to be 32.0% (min 13.3% in Liberia to max 50.0% in Nigeria). The determinants of late ANC visit were women’s age and education, partner’s education, health insurance coverage, birth order, household wealth, age at marriage, decision on health care, residence, and community-level women’s illiteracy.

My assessment of the manuscript is that it is well written and is well organized. I also see that this is the second reviewer of the manuscript which has improved after the authors addressed the previous reviewers’ comments. The findings from this study have the potential to impact health interventions and policy to improve initiation of timely ANC visit and maternal health services in developing countries.

Apart from few grammatical errors, which I have mentioned below and the authors need to address them, the manuscript can be considered for publication.

The authors may opt to use ANC visit instead of ANC booking, as currently they have used the two terms interchangeably, which can be confusing. Further, the women reported their ANC visit(s) rather than their booking of ANC visit(s).

The word “bias” is usually used as uncountable; therefore, in line 300 “may be recall bias” instead of “may be a recall bias” should be used.

In line 295 “This data was collected” to be changed to “The data were collected”.

The above are few errors I have noticed; the authors should check the English language and grammar for their work in this study.

7. PLOS authors have the option to publish the peer review history of their article (what does this mean?). If published, this will include your full peer review and any attached files.

Reviewer #2: No

Reviewer #4: No

---

## [Author Response · Author response to Decision Letter 2]

13 Dec 2024

Thank you again for the opportunity to revise our manuscript “Late initiation of antenatal care visit amid implementation of new antenatal care model in Sub-Saharan African countries: a multilevel analysis of multination population survey data (ID: PONE-D-24-21779R2)”. We have addressed the concerns raised by the reviewers using a point-by-point response as stated below. The amendments made to the manuscript have been presented using track change in the attachment titled “Revised Manuscript with Track Changes”.

Response to Editor’s comments

Journal Requirements:

Comment 1: Please review your reference list to ensure that it is complete and correct. If you have cited papers that have been retracted, please include the rationale for doing so in the manuscript text, or remove these references and replace them with relevant current references. Any changes to the reference list should be mentioned in the rebuttal letter that accompanies your revised manuscript. If you need to cite a retracted article, indicate the article’s retracted status in the References list and also include a citation and full reference for the retraction notice.

Response 1: Thank you. We have checked the reference list, confirmed that all references are complete and correct, and ensured that we have not used a retracted paper in the citation.

Additional Editor Comments:

Comment 1: Definition of the outcome – WHO recommends 1st ANC visit within 3 months (1st trimester). In the manuscript, the authors defined late ANC after 4 months of gestation. According to the literature, ANC visit within 4 months is also considered late ANC. Additionally, authors compared their results with papers mostly discussing late ANC after 3 months. I ask the authors to revisit WHO recommendations for ANC timing and revise their results accordingly. I also suggest the authors can consult and add the following citations which detail late ANC in another low-income country

• https://journals.plos.org/plosone/article?id=10.1371/journal.pone.0309300

• https://www.dovepress.com/sociodemographic-predictors-of-initiating-antenatal-care-visits-by-pre-peer-reviewed-fulltext-article-IJWH

Response 1: Thank you very much for your important and interesting recommendation. We have changed the definition for ANC timing, did a reanalysis, and revised the result section as per your suggestion. In addition, we have provided the suggested citations for the definition of late ANC visit.

Comment 2: In the Abstract add the abbreviation SSA from the first paragraph (line 25).

Response 2: Thank you. We have added the abbreviation for sub-Saharan Africa in line 25 of the abstract section.

Comment 3: In Abstract add the full form (DHS) from the first paragraph (line 27).

Response 3: Thank you have added the extended form of DHS in line 27 of the revised manuscript.

Comment 4: Methods described aren't the ones opted by the author. They were done part of the National Surveys. These methods should be supported be relevant citations (lines 94-100).

Response 4: Thank you very much. We have provided citation for DHS methodology as suggested in the revised manuscript.

Comment 5: Dependent variable: Add citation considering my earlier comment.

Response 5: Thank you. We have added the citations suggested in comment 1 for the dependent variable.

Response to Reviewer #4 Comments

Comment 1: My assessment of the manuscript is that it is well-written and is well organized. I also see that this is the second reviewer of the manuscript which has improved after the authors addressed the previous reviewers’ comments. The findings from this study have the potential to impact health interventions and policy to improve the initiation of timely ANC visits and maternal health services in developing countries. Apart from a few grammatical errors, which I have mentioned below and the authors need to address them, the manuscript can be considered for publication.

Response 1: Thank you for your appreciation and important concern. We have thoroughly checked and fixed the manuscript for grammatical and typos errors.

Comment 2: The authors may opt to use ANC visit instead of ANC booking, as currently, they have used the two terms interchangeably, which can be confusing. Further, the women reported their ANC visit(s) rather than their booking of ANC visit(s).

Response 2: Thank you for your suggestion. We have replaced “ANC booking” with “ANC visit” throughout the revised manuscript.

Comment 3: The word “bias” is usually used as uncountable; therefore, in line 300 “may be recall bias” instead of “may be a recall bias” should be used.

Response 3: Thank you for your concern. We have changed “may be a recall bias” to “may be recall bias” in the revised manuscript.

Comment 4: In line 295 “This data was collected” to be changed to “The data were collected”.

Response: Thank you very much. We have replaced “This data was collected” with “The data were collected” as suggested in the revised manuscript.

Comment 5: The above are a few errors I have noticed; the authors should check the English language and grammar for their work in this study.

Response 5: Thank you for your recommendation. We have checked and fixed grammatical and editorial errors throughout the revised manuscript.

---

## [Editor Report · Decision Letter 3]

16 Dec 2024

Late initiation of antenatal care visit amid implementation of new antenatal care model in Sub-Saharan African countries: a multilevel analysis of multination population survey data

PONE-D-24-21779R3

Dear Dr. Mare,

We’re pleased to inform you that your manuscript has been judged scientifically suitable for publication and will be formally accepted for publication once it meets all outstanding technical requirements.

Kind regards,

Muhammad Haroon Stanikzai

Academic Editor

PLOS ONE

Additional Editor Comments (optional):

Thank you for addressing the reviewer comments.

- Line 83: Please spell out SSA. Abbreviations should be spell out at their first use. You have spelled out in the abstract. You all also need to spell it out in the body of manuscript.

- Line 90: Please spell out DHS.

- Tables 1 and 4: Please correct Partner Education to Partner education.
---

## [Editor Report · Acceptance letter]

20 Dec 2024

PONE-D-24-21779R3 

PLOS ONE

Dear Dr. Mare, 

I'm pleased to inform you that your manuscript has been deemed suitable for publication in PLOS ONE. Congratulations! Your manuscript is now being handed over to our production team.

Kind regards, 

on behalf of

Dr. Muhammad Haroon Stanikzai 

Academic Editor

PLOS ONE